# Gender Analysis of Stress and Smoking Behavior: A Survey of Young Adults in Japan

Ayuka Yokoyama [1,*], Yuka Iwata [2], Nanami Oe [3] and Etsuko Tadaka [3,*]

1 Department of Nursing Informatics, Graduate School of Nursing Science, St. Luke's International University, Tokyo 104-0044, Japan
2 Department of Community Health Nursing, Graduate School of Medicine, Yokohama City University, Yokohama 236-0004, Japan; iwata.yuk.go@yokohama-cu.ac.jp
3 Department of Community and Public Health Nursing, Graduate School of Health Sciences, Hokkaido University, Sapporo 060-0812, Japan; o_nanami0706@pop.med.hokudai.ac.jp
* Correspondence: 22mn031@slcn.ac.jp (A.Y.); e_tadaka@pop.med.hokudai.ac.jp (E.T.)

**Abstract:** The global tobacco epidemic, claiming over 8 million lives annually, constitutes a formidable public health threat. Fatalities arise from both direct tobacco use and exposure to second-hand smoke. Smoking prevalence, notably in Japan, varies across age groups with distinct patterns indicating higher rates among those aged 40 years and above. Persistent concerns surround the significance of smoking behavior in individuals aged 20 to 30 years, given the potential for early adulthood behavior to contribute to long-term health impacts. The emergence of heated tobacco products adds complexity with a substantial percentage of individuals aged 20 to 30 years using these alternatives. This study analyzed data from 15,333 individuals aged 20 to 39 years, collected through Japan's "Comprehensive Survey of Living Conditions 2017". Compliant with the Japan Statistics Act, a secondary analysis employed multivariate logistic analysis to examine concerns and stress sources by sex and smoking behavior, adjusting for various variables. As a result, no statistically significant associations were found between smoking in men and concerns or stress. However, in women who smoked, significant associations were observed between smoking and specific stressors, such as work-related concerns, financial stress, and stress from a lack of personal free time. This study emphasizes the necessity of considering gender differences and social backgrounds in designing targeted smoking-prevention programs, aiming to enhance overall health longevity and comprehensively reduce lifelong smoking rates in this demographic.

**Keywords:** gender; stressors; smoking behavior; young adults; Japan





## 1. Introduction

The tobacco epidemic stands as one of the most formidable public health threats ever encountered globally, claiming the lives of over 8 million individuals each year. Of these fatalities, more than 7 million result from direct tobacco use. At the same time, an additional 1.3 million fatalities are attributed to non-smokers exposed to second-hand smoke (Institute for Health Metrics and Evaluation 2019). In 2020, 22.3% of the world's population (comprising 36.7% men and 7.8% women) was reported to be engaged in tobacco use (WHO 2023). Notably, Japan stands out among developed nations, owing to its high prevalence of smoking, revealing distinctive variations across age groups and genders.

Data released by the Japanese government for the year 2022 unveiled interesting patterns in smoking rates (MHLW 2022). The overall smoking rate was 25.4% for men and 7.7% for women. By age group, the smoking rate is as follows: 21.7% for men and 5.9% for women in the 20s age group, 29.9% for men and 9.0% for women in the 30s age group, 34.6% for men and 11.6% for women in the 40s age group, and 32.6% for men and 12.0% for women in the 40s age group. These statistics underscore a high prevalence of

smoking among individuals aged 40 years and above compared with the 20s and 30s age groups. However, concerns persist regarding the significance of smoking behavior among those in their 20s and 30s. This concern emanates from the potential for smoking behavior established in early adulthood to accumulate health impacts over a lifetime, contributing to a higher likelihood of persistent smoking (Bonnie et al. 2015). Smoking may be initiated in adolescence, not only in early adulthood. Adolescents may engage in smoking behaviors to enhance self-esteem, cope with stress, regulate weight, or as a buffer against negative emotions. These factors potentially increase the likelihood of adolescents continuing their smoking habits into adulthood (WHO 2024).

Young adults are known to use cigarettes and other tobacco products, such as electronic cigarettes (e-cigarettes), cigars, smokeless tobacco, and hookah (CDC 2022). Kinjo et al. (2020) and Tabuchi et al. (2019) reported that recent years had witnessed the proliferation of heated tobacco products, such as IQOS manufactured by Phillip Morris International, glo by British American Tobacco, and Ploom TECH by Japan Tobacco Inc. This introduces a new challenge in smoking behavior. The data indicated that approximately 16% of those in their 20s and 17% of those in their 30s utilize heated tobacco products (Odani and Tabuchi 2022). It is crucial to emphasize that all forms of tobacco use are harmful, and there is no safe level of exposure to tobacco. Encouraging a decline in smoking rates becomes not only a crucial step in improving individual health longevity for those in their 20s and 30s but also a pivotal measure in reducing overall societal smoking prevalence. Tjora et al. (2012) examined smoking initiation in early adulthood. Those who reported being smokers at age 30 years but not at age 18 years were defined as "late-onset smokers". The study indicated that early adulthood is a critical phase for daily smoking initiation (Tjora et al. 2012).

Although the association between smoking and stress has been highlighted in numerous previous studies (Lawless et al. 2015; Perski et al. 2022; Sato et al. 2022), Jahnel et al. (2019) described the process by which daily stress experiences linked social disadvantage and smoking behavior. Perski et al. (2022) revealed the prevalence of smoking to relieve stress. It associated smoking to relieve stress with motivation to quit but not significantly with actual attempts to quit smoking. Kim et al. (2019) examined the relationship between smoking cessation attempts and stress levels in South Korean smokers. The findings revealed that individuals who failed to quit smoking experienced higher stress than those who did not attempt smoking cessation, emphasizing the importance of addressing stress in smoking cessation interventions and health policies. However, there may be age and gender-specific variations in the association between smoking and stress (Tomioka et al. 2021). Nevertheless, there is still insufficient research specifically examining the relationship between smoking and stress in individuals aged 20 to 30 years, stratified by gender, in Japan. The smoking behavior of young adults may be influenced by the smoking behavior and attitudes of family members, friends, and others around smokers. A study of Korean middle and high school students found that family and friend smoking status was significantly associated with adolescent smoking behavior (Joung et al. 2016). One possible reason for this is that adolescents are susceptible to influence by social and environmental factors, such as home environment and school life (Park 2011).

Holliday and Gould (2016) indicated that gender differences may contribute to the relationship between stress and smoking behavior in adolescents. Therefore, stress may have different effects on smoking behavior for men and women. According to Byrne and Mazanov (1999, 2003), gender differences in smoking motivation emerge during adolescence. Adolescent females reported a higher incidence of smoking with higher perceived family and social stressors (Byrne and Mazanov 1999, 2003). Consideration of these gender differences in the context of stress and smoking is deemed necessary. To decrease smoking rates among young adults aged 20 to 30 years, it is imperative to understand how stressors identified by gender exert influence on smoking behavior. In this age group, the impact of individual stress experiences on smoking behavior is diverse and intricate, making understanding crucial for the development of effective smoking-prevention programs (Holliday and Gould 2016). Identifying stress factors specific to each

gender is essential, considering the influence of gender differences and social backgrounds. Graves et al. (2021) revealed gender differences in stress perception and coping mechanisms among college students in the United States. They found that female students experienced higher levels of perceived stress compared with their male counterparts. Additionally, the study highlighted that female students tended to employ more emotion-focused coping strategies, such as seeking social support and self-distraction (Graves et al. 2021). These studies revealed that stress factors and coping mechanisms differ based on gender. The remaining challenge is to elucidate how specific stressors identified by gender impact smoking behavior as a coping mechanism.

The developmental challenges in the 20–30 year age group encompass the pursuit of economic self-sufficiency, vocational advancement, the establishment of relationships with family and friends, and the exploration of self-identity and hobbies. Aligned with Havighurst's lifespan developmental tasks (Havighurst 1972), this period corresponds to the lifelong developmental challenges of 'Occupational Attainment' and 'Social Integration'. Economic independence and professional growth are intimately connected with occupational adaptation, while building relationships with family and friends is integral to social integration. Concerns related to work, economic pressures, and stress in familial or non-familial relationships significantly impact the achievement of these lifelong developmental goals. Simultaneously, stress stemming from limited free time may influence efforts toward self-understanding and personal growth. Individuals during this phase must adeptly navigate and balance these developmental tasks, as highlighted by Havighurst (1972), to construct a fulfilling life and foster robust health and development. Identifying stress factors specific to gender in individuals aged 20 to 30 years and understanding their impact on smoking behavior are crucial for the development of more effective and targeted smoking-cessation strategies. This approach holds the potential to not only improve the overall health longevity of individuals in the 20–30 year age group but also comprehensively reduce lifelong smoking rates in future society. Healthy longevity is defined as "the state in which years in good health approach the biological lifespan with physical, cognitive, and social functioning, enabling well-being across populations" (National Academy of Medicine 2022). Therefore, the primary objective of this study is to elucidate the relationship between stress factors and smoking behavior with a focus on gender by using a large-scale government dataset that represents the youth population aged 20 to 30 years in Japan, utilizing a Comprehensive National Survey dataset.

## 2. Materials and Methods

### 2.1. Data Source and Study Design

The Comprehensive Survey of Living Conditions in Japan targets households randomly selected nationwide by the government. The survey's objective is to capture public statistics on health, medical care, welfare, pensions, income, and various aspects of life, providing a foundation for the formulation and implementation of health and labor policies. The survey is conducted every three years using a combination of mail and face-to-face interviews conducted by dedicated national surveyors. This study constitutes a secondary analysis conducted following the Japan Statistics Act (Act No. 18 of 1947), limited to the purposes of public interest and academia.

The data for this study were derived from 16,101 individuals aged 20 years to under 40 years who participated in the "Comprehensive Survey of Living Conditions 2017" by the Ministry of Health, Labour and Welfare in Japan. Of these individuals, 768 people were excluded because they did not answer questions regarding smoking behavior and experience of stress. Thus, the final sample for the analysis comprised 15,333 individuals (effective response rate: 95.2%). All respondents completed the entire survey.

#### 2.1.1. Demographic Characteristics

The demographic information encompassed age, gender, marital status, educational level, employment status, and self-reported health status. Employment status was deter-

mined using an actual method, categorizing individuals as 'Employed' if they engaged in any income-generating work or academic pursuits during the one-month survey period and 'Unemployed' if they did not. Health status was evaluated on a 5-point Likert scale: "Excellent", "Good", "Fair", "Poor", and "Very Poor".

### 2.1.2. Dependent Variables

Regarding the current smoking behavior of the participants as a dependent variable, this study defined individuals as "Smokers" if they endorsed "Smoke every day" or "Smoke occasionally" and as "Non-smokers" if they reported "Used to smoke but currently do not" or "Do not smoke". Additionally, the number of cigarettes smoked per day was recorded for smokers.

### 2.1.3. Independent Variables

Regarding stress as an independent variable, based on Havighurst's lifespan developmental tasks for individuals aged 20 to 30 years (Havighurst 1972), stressors considered crucial for the study population were selected from the "Comprehensive Survey of Living Conditions 2017" (MHLW 2017). These stressors, encompassing both variables and definitions, included (a) work-related concerns and stress; (b) financial concerns and stress (income, household finances, and debt); (c) family relationship strains (stress from familial relationships); (d) non-familial interpersonal struggles (stressors in non-familial relationships); (e) stress from lack of personal free time (challenges due to limited leisure time); sense of purpose concerns (struggles in finding life purpose); and (f) others (other concerns and stressors).

### 2.2. Statistical Analysis

This study conducted descriptive analyses of demographic characteristics, smoking behavior, stress, and stress coping, all stratified by gender. Following univariate analysis, we employed a binary logistic regression model to calculate odds ratios (OR) and 95% confidence intervals (CI) for smoking and non-smoking across different stress and stress-coping categories. Notably, we considered age, educational level, marital status, employment status, and health status as confounding factors significantly associated with stress and smoking behavior. These factors were incorporated into the model to enhance the accuracy of our results. Statistical significance was determined by a *p*-value < 0.05 or a 95% confidence interval (CI) that did not include 1. The analysis was conducted using the SPSS software package (version26.0; IBM Corp., Armonk, NY, USA).

### 2.3. Ethical Approval

This research adhered to the principles outlined in the 1964 Declaration of Helsinki (and its subsequent amendments) and followed the ethical guidelines for life sciences and medical research involving human subjects provided by the Ministry of Health, Labour and Welfare of Japan. Approval was obtained from the Institutional Review Board of the School of Health Sciences, Hokkaido University (No. 23–67, 2 October 2023). This study, conducted under approval (No. 23002) from the Minister of Health, Labour and Welfare, was carried out under Japanese statistical laws. Survey information was processed to ensure non-identifiability of specific individuals, corporations, or entities, and this study was considered to significantly contribute to the advancement of academic research and the creation of statistics with substantial public interest.

## 3. Results

### 3.1. Demographic Characteristics of Participants by Sex and Smoking Status

Table 1 presents the results of the demographic characteristics of the participants, categorized by sex and smoking behavior. The percentage of smokers was 37.4% among men (*n* = 2828) and 11.5% among women (*n* = 891). Both men and women showed a statistically significant increase in smoking rates with higher age categories, peaking at

35–39 years ($p < 0.001$). Additionally, smoking rates for both sexes were significantly higher with lower educational levels ($p < 0.001$) and worsening health conditions ($p < 0.001$).

**Table 1.** Demographic characteristics of participants by sex and smoking behavior ($n = 15,333$).

| | Men, *n* = 7553 | | | | | Women, *n* = 7780 | | | | |
| --- | --- | --- | --- | --- | --- | --- | --- | --- | --- | --- |
| | Smoker *n* = 2828 | | Non-Smoker *n* = 4725 | | | Smoker *n* = 891 | | Non-Smoker *n* = 6889 | | |
| | *n* | % | *n* | % | *p** | *n* | % | *n* | % | *p** |
| Age (years) | | | | | <0.001 | | | | | <0.001 |
| 20–24 | 418 | 14.8 | 1088 | 23.0 | | 120 | 13.5 | 1390 | 20.2 | |
| 25–29 | 578 | 20.4 | 1042 | 22.1 | | 207 | 23.2 | 1463 | 21.2 | |
| 30–34 | 811 | 28.7 | 1185 | 25.1 | | 248 | 27.8 | 1831 | 26.6 | |
| 35–39 | 1021 | 36.1 | 1410 | 29.8 | | 316 | 35.5 | 2205 | 32.0 | |
| Educational level | | | | | <0.001 | | | | | <0.001 |
| Junior high school | 184 | 6.5 | 132 | 2.8 | | 90 | 10.1 | 159 | 2.3 | |
| High school | 1092 | 38.6 | 1181 | 25.0 | | 354 | 39.7 | 1599 | 23.2 | |
| Professional Training College | 342 | 12.1 | 602 | 12.7 | | 155 | 17.4 | 1080 | 15.7 | |
| Junior college | 73 | 2.6 | 122 | 2.6 | | 63 | 7.1 | 995 | 14.4 | |
| University | 691 | 24.4 | 1907 | 40.4 | | 72 | 8.1 | 2188 | 31.8 | |
| Graduate school | 79 | 2.8 | 314 | 6.6 | | 4 | 0.4 | 147 | 2.1 | |
| Missing data | 367 | 13.0 | 467 | 9.9 | | 153 | 17.2 | 721 | 10.5 | |
| Marital status | | | | | <0.001 | | | | | 0.622 |
| Married | 1392 | 49.2 | 1878 | 39.7 | | 434 | 48.7 | 3416 | 49.6 | |
| Unmarried | 1436 | 50.8 | 2847 | 60.3 | | 457 | 51.3 | 3473 | 50.4 | |
| Working status | | | | | 0.001 | | | | | 0.089 |
| Employed | 2687 | 95.0 | 4419 | 93.5 | | 841 | 94.4 | 6605 | 95.9 | |
| Unemployed | 115 | 4.1 | 279 | 5.9 | | 42 | 4.7 | 247 | 3.6 | |
| Missing data | 26 | 0.9 | 27 | 0.6 | | 8 | 0.9 | 37 | 0.5 | |
| Health status | | | | | <0.001 | | | | | <0.001 |
| Excellent | 679 | 24.0 | 1338 | 28.3 | | 154 | 17.3 | 1640 | 23.8 | |
| Good | 524 | 18.5 | 952 | 20.1 | | 185 | 20.8 | 1418 | 20.6 | |
| Fair | 1417 | 50.1 | 2077 | 44.0 | | 429 | 48.1 | 3171 | 46.0 | |
| Poor | 174 | 6.2 | 291 | 6.2 | | 103 | 11.6 | 575 | 8.3 | |
| Very poor | 26 | 0.9 | 47 | 1.0 | | 15 | 1.7 | 47 | 0.7 | |
| Missing data | 8 | 0.3 | 20 | 0.4 | | 5 | 0.6 | 38 | 0.6 | |
| Number of cigarettes smoked/day | | | | | | | | | | |
| Less than 10 cigarettes | 1049 | 37.1 | - | - | | 449 | 50.4 | - | - | |
| 11–20 cigarettes | 1399 | 49.5 | - | - | | 386 | 43.3 | - | - | |
| 21–30 cigarettes | 297 | 10.5 | - | - | | 46 | 5.2 | - | - | |
| 31 cigarettes or more | 44 | 1.6 | - | - | | - | - | - | - | |
| Missing data | 39 | 1.4 | - | - | | 10 | 1.1 | - | - | |

* Chi-square test.

*3.2. Univariate Analysis on Sources of Concerns and Stress by Sex and Smoking Behavior*

Table 2 presents the results of a univariate analysis on sources of stress for men and women. Smoking in men showed statistically significant associations only with Financial Concerns and Stress: Income ($p = 0.001$) and Other Concerns and Stressors ($p = 0.007$). On the other hand, smoking in women was significantly associated with Work-related Concerns and Stress ($p = 0.031$), Financial Concerns and Stress ($p < 0.001$), Family Relationship Strains ($p = 0.002$), Non-Familial Interpersonal Struggles ($p = 0.022$), and Lack of Personal Free Time Stress ($p = 0.001$).

**Table 2.** Univariate analysis on sources of stress by sex and smoking behavior (*n* = 15,333).

| | Men, *n* = 7553 | | | | | Women, *n* = 7780 | | | | |
| | Smoker *n* = 2828 | | Non-Smoker *n* = 4725 | | | Smoker *n* = 891 | | Non-Smoker *n* = 6889 | | |
| | *n* | % | *n* | % | *p* * | *n* | % | *n* | % | *p* * |
|---|---|---|---|---|---|---|---|---|---|---|
| Work-related Concerns and Stress | 823 | 29.1 | 1358 | 28.7 | 0.738 | 265 | 29.7 | 1815 | 26.3 | 0.031 |
| Financial Concerns and Stress: Income, household finances, and debt-related worries | 447 | 15.8 | 612 | 13.0 | 0.001 | 263 | 29.5 | 1140 | 16.5 | <0.001 |
| Family Relationship Strains: Stress from familial relationships | 131 | 4.6 | 202 | 4.3 | 0.464 | 97 | 10.9 | 541 | 7.9 | 0.002 |
| Non-Familial Interpersonal Struggles: Stressors in non-familial relationships | 210 | 7.4 | 347 | 7.3 | 0.895 | 113 | 12.7 | 702 | 10.2 | 0.022 |
| Lack of Personal Free Time Stress: Challenges due to limited leisure time | 156 | 5.5 | 275 | 5.8 | 0.582 | 97 | 10.9 | 531 | 7.7 | 0.001 |
| Sense of Purpose Concerns: Struggles in finding life purpose | 159 | 5.6 | 273 | 5.8 | 0.778 | 54 | 6.1 | 320 | 4.6 | 0.063 |
| Others: Other Concerns and Stressors | 68 | 2.4 | 166 | 3.5 | 0.007 | 23 | 2.6 | 233 | 3.4 | 0.207 |

* Chi-square test.

### 3.3. Multivariate Logistic Analysis on Sources of Concerns and Stress by Sex and Smoking Behavior

Table 3 presents the results of a multivariate logistic analysis on sources of concerns and stress for men and women, adjusting for age, educational level, marital status, working status, and health status. No statistically significant associations were observed between smoking in men and all types of stress. However, smoking in women showed significant associations with Work-related Concerns and Stress (OR = 1.254, 95% CI = 1.047–1.502), Financial Concerns and Stress (OR = 1.639, 95% CI = 1.363–1.970), and Lack of Personal Free Time Stress (OR = 1.393, 95% CI = 1.063–1.825).

**Table 3.** Multivariate logistic analysis on sources of stress by sex and smoking behavior (*n* = 15,333).

| | Men | | | Women | | |
| | *p* | OR | 95% CI | *p* | OR | 95% CI |
|---|---|---|---|---|---|---|
| Age [a] | <0.001 | 1.219 | 1.168–1.272 | <0.001 | 1.130 | 1.060–1.205 |
| Educational level [a] (Graduate school = 1~junior high school = 6) | <0.001 | 1.370 | 1.324–1.417 | <0.001 | 1.919 | 1.793–2.055 |
| Marital status [a] (unmarried = 1, married = 2) | <0.001 | 1.470 | 1.338–1.614 | 0.622 | 0.966 | 0.840–1.110 |
| Working status [a] (unemployed = 1, employed = 2) | 0.001 | 1.475 | 1.181–1.843 | 0.090 | 0.749 | 0.536–1.047 |
| Health status [a] (excellent = 1~very poor = 5) | <0.001 | 1.114 | 1.062–1.169 | <0.001 | 1.219 | 1.132–1.313 |
| Work-related Concerns and Stress [b] (0/1) | 0.865 | 0.990 | 0.881–1.113 | 0.014 | 1.254 | 1.047–1.502 |
| Financial Concerns and Stress: Income, household finances, and debt [b] (0/1) | 0.214 | 1.098 | 0.947–1.273 | <0.001 | 1.639 | 1.363–1.970 |
| Family Relationship Strains: Stress from familial relationships [b] (0/1) | 0.785 | 0.966 | 0.754–1.238 | 0.656 | 1.062 | 0.815–1.384 |
| Non-Familial Interpersonal Struggles: Stressors in non-familial relationships [b] (0/1) | 0.634 | 0.953 | 0.784–1.160 | 0.253 | 1.151 | 0.905–1.463 |
| Lack of Personal Free Time Stress: Challenges due to limited leisure time [b] (0/1) | 0.400 | 0.908 | 0.726–1.136 | 0.016 | 1.393 | 1.063–1.825 |

[a] Univariate analysis. [b] Multivariate logistic analysis adjusting for age, educational level, marital status, working status, and health status.

## 4. Discussion

First, this study underscores gender disparities in the correlation between smoking behavior and potential stressors contributing to smoking among individuals aged 20 to 30 years. Notably, there were no statistically significant associations observed between smoking and stressors in men. In contrast, significant associations were identified in women, particularly regarding concerns related to work, financial stress, and a lack of personal free time. When comparing these findings with a prior study (Sato et al. 2022), we observed partial alignment, possibly stemming from variations in the definition of smokers. Although the earlier study defined smokers as 'daily smokers', our current study has broadened the definition to include both 'daily smokers and occasional smokers', revealing distinct gender differences in the smoking–stressor association, including occasional smokers.

Second, this study brings attention to gender differences in the relationship between smoking and stressors among young adults aged 20 to 30 years, manifesting as early as the onset of smoking in the 20s. Results of this study showed that triggers for smoking initiation during the 20s may vary across different life stages, suggesting that concerns related to work, financial stress, and a lack of personal free time could act as catalysts for women. Smoking behavior initiated in the 20s, whether daily or occasional, holds the potential for lifelong persistence, contributing to heightened health risks due to the extended duration of such behavior. The initiation of smoking in the 20s poses significant public health concerns nationwide. In summary, this study elucidates gender-specific stressors as triggers for smoking initiation in the 20s, emphasizing these stressors as potential long-term health risks associated with smoking behavior initiated during this critical period.

### 4.1. Smoking Prevalence and Sources of Concerns and Stress by Gender

This study suggests that smoking rates differed significantly between men and women. It indicates a significant difference in smoking prevalence between men (37.4%) and women (11.5%), aligning with national trends where smoking rates among men are notably higher than those among women in Japan. Moreover, our findings reveal an increase in smoking rates with age and a correlation between lower educational levels and worsening health conditions. In 2021, the smoking rates among general adults in the continental United States were 13.1% for men, 10.1% for women, 12.9% for non-Hispanic White adults, 11.7% for non-Hispanic Black adults, and 5.4% for non-Hispanic Asian adults (CDC 2023). In comparison, the smoking rates among Japanese individuals in the 20–30 year age group in this study, particularly among men, are exceptionally high. These findings underscore the importance of formulating smoking-prevention strategies that take into account gender differences among the target population of individuals in the 20–30 year age group. Table 2 provides insights into the sources of concerns and stressors, indicating divergent patterns between men and women. Financial concerns and stress related to income were significantly associated with smoking in men, suggesting that economic factors may contribute significantly to smoking behavior in this group. On the other hand, women showed associations with a broader range of stressors, including work-related concerns, family relationship strains, non-familial interpersonal struggles, and lack of personal free time stress. This finding is consistent with prior research (Graves et al. 2021; Viertiö et al. 2021) indicating gender differences in the perception of stress. However, the evidence supporting the impact of stress on the smoking behavior of women is not sufficient, and its generation is indeed a subject for future research. Moreover, this finding also suggests a more multifaceted interplay of stressors influencing smoking behavior in women.

### 4.2. Multivariate Analysis of Stressors and Smoking Behavior and Further Implications

In this study, significant gender-based differences in stressors influencing smoking behavior were observed. The analysis did not reveal statistically significant associations between smoking in men and specific concerns or stressors. In contrast, women exhibited significant associations with work-related concerns, financial stress, and stress from lack of personal free time. These findings underscore the importance of comprehensively con-

sidering stressors, particularly for women, when developing targeted smoking cessation programs. To further contextualize the observations in our study within Havighurst's developmental tasks during adolescence, it is crucial to recognize that young individuals in this stage engage in various tasks related to personal and social development, including identity establishment, interpersonal relationship development, and the pursuit of autonomy. Considering these developmental tasks can provide insights into the unique stressors that influence smoking behavior in young adults. The stressors identified in women may be linked to challenges in establishing a stable identity and managing interpersonal relationships. Recognizing these connections allows for the development of intervention strategies addressing the underlying developmental tasks, providing support in navigating the complexities of young adulthood in women. However, the initial observation regarding the significance of workplace competition and financial stress for men may require reconsideration, as the data did not show statistically significant associations for men. The stressors that influence smoking behavior in men may be different from those in women and require further exploration.

Furthermore, this study suggests the importance of considering the developmental tasks and gender differences faced by adolescents when devising intervention strategies. Tate et al. (2021) explored socio-environmental and psychosocial factors influencing the susceptibility of adolescents to smoking, emphasizing the role of diverse sociocultural characteristics in shaping smoking behaviors. Therefore, for example, in some societies, smoking is more accepted among men than women, often reinforced by a masculine image in advertising and media. For instance, interventions for young men could focus on identity development and psychosocial expectations associated with smoking. Conversely, interventions for young women could focus on identity development and stressors related to individual, family, and social expectations.

There are creative smoking cessation programs that would benefit from this study's findings. For example, Curran et al. (2023) reported that a digital media mental health literacy campaign called "What's up with everyone?" improved mental health literacy among young people in the United Kingdom. The animations were created using "Health Humanities," a new field of study that has recently been gaining momentum around the world. Health Humanities "brings those in the arts and humanities together with those in the health care fields to work collaboratively in the domains of education, research, and practice toward new insights, innovations, and activities that result in improved health and well-being of patients, healthcare professionals themselves, informal carers, and society as a whole" (Huffman and Inoue 2019). The development of comprehensive health humanities programs to improve young people's ability to cope with stress, such as mental health literacy, could contribute to the prevention of smoking.

### 4.3. Limitations and Future Directions

This study acknowledges its limitations, including the reliance on self-reported data and the cross-sectional nature of the analysis, preventing the assertion of a causal relationship. Another limitation is that it does not consider comorbid mental health conditions and the role this may play in smoking, as well as stress and coping. A further shortcoming is the lack of information on tobacco products other than cigarettes used by the respondents, their reasons for starting to smoke, and their smoking history. Recognizing these constraints is crucial for a comprehensive interpretation of our findings. To enhance the understanding of gender differences in stressors and smoking behavior among youth, future research should employ longitudinal studies.

### 5. Conclusions

This study highlights significant gender differences in smoking behavior among young adults. No statistically significant associations were found between smoking in men and concerns or stress. However, smoking in women showed strong associations with work-related concerns, economic stress, and a lack of personal freedom. These

results underscore the importance of addressing the multifaceted nature of stressors when designing effective smoking cessation programs, especially focusing on young women. Furthermore, the results emphasize the need for a review of comprehensive programs that will improve mental health literacy, including stress-coping ability in young men. In conclusion, incorporating these insights will significantly contribute to understanding the complexity of smoking behavior in young individuals and the development of targeted strategies during this critical period for lifelong smoking.

**Author Contributions:** Conceptualization: A.Y., Y.I. and E.T.; methodology, N.O.; software, N.O.; validation, A.Y., Y.I., N.O. and E.T.; formal analysis, N.O.; investigation, E.T.; resources, E.T.; data curation, A.Y., Y.I., N.O. and E.T.; writing—original draft preparation, A.Y.; writing—review and editing, A.Y., Y.I., N.O. and E.T.; visualization, Y.I. and N.O.; supervision, E.T.; project administration, E.T.; funding acquisition, E.T. All authors have read and agreed to the published version of the manuscript.

**Funding:** The sources of funding for our study were Grants-in-Aid for Scientific Research (KA-KENHI) of the Japan Society for the Promotion of Science Grant Number 23H0321903 (PI: Etsuko TADAKA).

**Institutional Review Board Statement:** The study was conducted under the Declaration of Helsinki and was approved by the Institutional Review Board of the School of Health Sciences, Hokkaido University (Protocol No. 23-67; 28 September 2023). Additionally, it received approval from the Japan Minister of Health, Labour, and Welfare (Protocol No. 23002; 11 July 2023) in compliance with the Japanese Statistical Laws.

**Informed Consent Statement:** All participants in this study provided written informed consent, affirming their voluntary participation rights or the right to decline, secured through the assurance of complete anonymity. Additionally, written informed consent was obtained for the disclosure of the survey results.

**Data Availability Statement:** The data utilized in this study emanate from the National Survey which is under the custodian-ship of the Cabinet Office of Japan. Access to the comprehensive dataset is contingent upon adherence to the procedures outlined in the Statistics Act of Japan. Prospective users are required to undergo an application process, seeking approval for data access, under the statutory man-dates. Permission is granted solely under circumstances where the proposed utilization of the data aligns with the public interest, such as the generation of new statistics or contributions to academic research. For detailed information on the application procedures, please refer to the Statistics Bureau of Japan website at https://www.stat.go.jp/english/index.html (accessed on 27 January 2024).

**Acknowledgments:** The authors express their gratitude to all individuals who gave their time and energy to participate in this study.

**Conflicts of Interest:** The authors declare no conflicts of interest. The funders had no role in the design of the study; in the collection, analyses, or interpretation of data; in the writing of the manuscript; or in the decision to publish the results.

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
