# Peer review of "Gender Analysis of Stress and Smoking Behavior: A Survey of Young Adults in Japan"

_socsci, doi:10.3390/socsci13030128_

Round 1

Reviewer 1 Report

Comments and Suggestions for Authors

Thank you for the opportunity to review this interesting and valuable addition to the literature. Please see below for my detailed feedback and suggestions.

Title:

The title of the article can be more concise. For example, “Analysis of a survey of young adults in Japan” could be “A survey of young adults in Japan”. I am also not certain I would refer to “gender-specific stressors” given that the same stressors were examined across genders. Consider phrases such as “a gender analysis”, “a gendered analysis”, or “a gendered perspective” and so on.

Abstract

The abstract provides a good overview of the article.

1. Introduction

The Introduction addresses most key points but can be more cohesive and succinct. Please also consider that:

·        Smoking may be initiated in adolescence, not only in early adulthood.

o   What are some of the dynamics around later-onset smoking? Focusing more on the dynamics of later-onset smoking can also help set the stage for another aspect to revisit in your Discussion.

·        Stress may not be the only reason for initiating and maintaining smoking

o   What about the influence of friends and peers?

o   What about those who came from households where family members (particularly parents) smoked?

·        While I appreciated the addition of a section on heated tobacco products on pages 1 - 2 (“Recent years have witnessed the proliferation of heated tobacco products, including 44 IQOS (Phillip Morris International, Neuchatel, Switzerland), glo (British American To-45 bacco, London, UK), and Ploom TECH (Japan Tobacco Inc., Tokyo, Japan), introducing a new challenge in smoking behavior.”), you do not discuss any specific types of products used by participants in your results, as you specifically discuss cigarettes.

o   Is it a further shortcoming of your study that you did not consider other products?

o   What is known about the use of tobacco products other than cigarettes in the age groups of interest in the study?

·        For the in-text reference “According to the World Health Organization (WHO) Fact Sheet for 2020”, you can simply state “According to the World Health Organization (2020)”, or “According to the WHO (2020)”. Please also check the link in the reference list.

I have no further input on the Materials and Methods and Results sections.

4. Discussion

Some aspects of the Discussion are presented like a Results section:

4.1. Smoking Prevalence and Sources of Concerns and Stress by Gender

Table 1 indicates a significant difference in smoking prevalence between ...”

“Table 3, Multivariate Analysis on Stressors and Smoking Behavior, refined …”

·        Statements such as these should be integrated better. “We found a significant difference in smoking prevalence between …” and “Further multivariate analysis on stressors and smoking behavior revealed …”

The figures you cite here are very dated: “According to data from the American Lung Association, the smoking rates among general adults in the continental United States are approximately 15.6% for men, 12.0% for women, 15.0% for White individuals, 14.6% for Black individuals, and 7.4% for Asian individuals (Allen et al. 2014).”

·        More recent figures from the USA are available here: https://www.cdc.gov/tobacco/data_statistics/fact_sheets/adult_data/cig_smoking/index.htm

At the outset of the paper you refer to Havighurst’s theory but this is not well-integrated into the Discussion. Further to that, there is a need to address the implications of the study concretely.

·        Despite making frequent reference to the need for prevention strategies and that these should be gender-specific, no specific implications for interventions are outlined.

·        It would be helpful to explore existing strategies in the study context and to look to the literature for potential approaches.

·        A way to address both concerns I raise there would be to frame your recommendations within the aforementioned theory.

Limitations and Future Directions

·        You indicate that extrapolating your findings from adults aged 20 to 30 years to other age groups should be done cautiously however your study is premised on the link between smoking in this age group, and smoking at a later stage (ages 40 to 50).

o   Would it not be more reasonable to consider that (1) not all smokers in their 20s and 30s will continue smoking into their 40s and 50s and that (2) not all smoking in the latter age group would have had a late onset? What about individuals who start smoking early, such as during adolescence?

o   You should consider that you do not have information on why respondents initiated smoking, or how long they have been smoking.

·        You also state that future research should explore longitudinal data to better understand the temporal dynamics of stress and smoking behaviour. Given that the study is conducted every three years, I wondered about the possibility of suggesting that participants’ responses should be linked across study waves.

·        Another limitation is that you do not consider comorbid mental health conditions and the role this may play in smoking, and in stress and coping.

Conclusion

·        In the conclusion you restate the need for targeted interventions, although the article itself did not offer any recommendations. I would revise the conclusion to better provide closure to the study and situate the study’s significance.

Reviewer 2 Report

Comments and Suggestions for Authors

 Correlation between Gender-Specific Stressors and Smoking 2 Behavior: Analysis of a Survey of Young Adults in Japan

Review

Overall

Thank you for the opportunity to review the manuscript. This is an important topic and relevant across different population groups. There are some comments and suggestions that the authors could work on as they develop to improve the manuscript. They could explain the reason for choosing the target age-group because at the end they indicate it is a limitation; the introduction needs to be strengthened with additional references and clear definition and explanation of concepts that are introduced.

The methods section could benefit from having definitions of all the variables that they use in the analysis, results, and discussion.

The results need to be clearly structured. For example they could present results about men and women and then clearly present the overall results. That might help with the flow of the presentation of results.

The discussion needs additional references to support some of the statements presented. In addition, it is important to be cautious to avoid presenting results as a causation when it is correlational relationship.

Additional and specific comments are below;

Introduction

Page 1 line 33 - 32

 Analysis of the 2022 data from the Japanese government unveils intriguing patterns 33 in smoking prevalence.

What is the overall smoking prevalence from the Japanese government data?

Page 1 line 34-37

 Among men aged 20 to 30 years, smoking rates range from 21.7% 34 to 29.9%, whereas for women in the same age group, the rates fluctuate between 5.9% and 35 9.0%. In contrast, men aged 40 to 50 years exhibit rates ranging from 32.6% to 34.6%, and 36 women in the same age bracket have rates between 11.6% and 12.0%.

what do you mean by "the rate fluctuate between 5.9% and 9.0%"?

The range is not clear - from what age group  to what age group?

Page 1 line 38 – 39

These statistics underscore the highest 38 prevalence of smoking among individuals aged 40 years and above.

It is not clear what you are comparing age group 40 and above.

Page 1 lines 39-42

 However, concerns 39 persist regarding the significance of smoking behavior among those in their 20s and 30s. 40 This concern emanates from the potential for smoking behavior established in early adult-41 hood to accumulate health impacts over a lifetime, contributing to a higher likelihood of persistent smoking.

Where are these concerns coming from - could you provide references to show these concerns?

Page 1 line 44

 Recent years have witnessed the proliferation of heated tobacco products, including IQOS (Phillip Morris International, Neuchatel, Switzerland), glo (British American Tobacco, London, UK), and Ploom TECH (Japan Tobacco Inc., Tokyo, Japan), introducing a

Please provide reference/s.

Page 2 lines 47-48

Data indicates that approximately 40% of men and 47 50% of women in their 20s to 30s are utilizing these products (Odani and Tabuchi 2020).

Could you be more specific - it 20-year-old to 30 year older.

Page 2 lines 53 – 54

Although the association between smoking and stress has been highlighted in numerous previous studies, …

Please provide reference/s

Page 2 lines 55 – 57

Perski et al. have explored smoking and stress relief, revealing prevalent stress-related smoking. It associates stress-related smoking with motivation to quit but not significantly with actual quit attempts

Define “smoking related stress”

Page 2 lines 62 – 63

However, there may be age and gender-specific variations in the association between 62 smoking and stress.

Provide reference/s

Page 2 lines 63 – 64

Yet, there is still insufficient research specifically examining the rela-63 tionship between smoking and stress in individuals aged 20 to 30, stratified by gender.

What is the basis for this population group?

Page 2 lines 65 – 66

To decrease smoking rates among young adults aged 20 to 30, it is imperative to un-65 derstand how stressors identified by gender exert influence on smoking behavior.

Clarify this statement

Page 2 lines 66 – 68

In this 66 age group, the impact of individual stress experiences on smoking behavior is diverse and 67 intricate, making understanding crucial for the development of effective smoking prevention programs.

Provide the basis for this - e.g. references to studies that have concluded so.

Page 2 lines 70 – 71

The study by Graves et al. 70 examined gender differences in stress perception and coping mechanisms among college 71 students.

Where and when was the study conducted?

Page2 lines 76 – 77

The study by Johansen et al. focused on 76 mental distress among young adults, highlighting the impact of gender differences and 77 social support.

Define mental distress and stress - are they two different concepts

Page 2 lines 78 – 79

The research, conducted in Norway, revealed that social support signifi-78 cantly reduces mental distress, particularly in young women

Do you have studies that examine stress and smoking behavior compared between males and females. The Johnson study does not examine the smoking aspect of stress but just examine the comparison between males and females on stress.

Page 2 lines 80 – 81

The re-80 maining challenge is to elucidate how specific stressors identified by gender impact smoking behavior as a coping mechanism.

Could you define specific stressors?

Page 2 line 80

The developmental challenges in the 20s to 30s encompass the pursuit of economic

You could use 20-30 age-group as 20s and 30s are not standard terminologies.

Page 2 lines 85 – 87

Aligned with Havighurst's 85 lifespan developmental tasks (Havighurst 1948), this period corresponds to the lifelong 86 developmental challenges of 'Occupational Attainment' and 'Social Integration.

Do you have more recent references?

Page 2 lines 87 - 92

Economic independence and professional growth are intimately connected with occupational adap-88 tation while building relationships with family and friends is integral to social integration. 89 Concerns related to work, economic pressures, and stress in familial or non-familial rela-90 tionships significantly impact the achievement of these lifelong developmental goals. Sim-91 ultaneously, stress stemming from limited free time may influence efforts toward self-92 understanding and personal growth. Individuals

Page 2 lines 98 – 99

This approach holds the potential to not only improve the 98 overall health longevity of individuals in their 20s to 30s but also comprehensively reduce.

Define health longevity.

2. Materials and Methods

Page 3 line 142

2.3. Statistical Analysis

Do you apply sampling weights to the analysis?

What was the statistical test/s used?

3. Results

Page 5 line 178

Table 2. Univariate analysis on sources of stress by sex and smoking behavior (n = 15,333).

All the variables in the analysis need to be defined in the methods section – for example how was “smoker” measured and how was “Work-related concerns and stress” measured.

Page 5 line 182 – 183

 No statistically significant associations were ob-182 served between smoking in men and any stress.

How did you measure or calculate “any stress”?

4. Discussion

Page 6 lines 203 - 205

Specifically, triggers for smoking initiation during 203 the 20s may vary across different life stages, suggesting that concerns related to work, 204 financial stress, and a lack of personal free time could act as catalysts for women. Smoking

Provide reference/s

Page 7 lines 215 – 216

Moreover, our findings reveal an in-215 crease in smoking rates with age and a correlation between lower educational levels and 216 worsening health conditions.

You did not measure education in the study.

Page 217 – 220

According to data from the American Lung Association, the 217 smoking rates among general adults in the continental United States are approximately 218 15.6% for men, 12.0% for women, 15.0% for White individuals, 14.6% for Black individu-219 als, and 7.4% for Asian individuals.

You could use more recent estimates.

Page 7 lines 220 – 221

In comparison, the smoking rates 220 among Japanese individuals in their 20s to 30s in this study, particularly among men, are 221 exceptionally high

These are two different population groups in different geographic location and different age groups

Page 7 lines 222 – 223

These findings underscore the importance of formulating smoking 222 prevention strategies that take into account gender differences among the target popula-223 tion of individuals in their 20s to 30s.

This is not a strong argument - it is not supported by the data.

Page 7 lines 225-227

Financial concerns 225 and stress related to income were notably associated with smoking in men, underlining 226 economic factors as significant contributors to smoking behavior in this group.

This should be “correlated and no causation”

Page 7 lines 232-236

The results indicate that women tend to report 232 higher levels of stress compared to men, and there are also discernible gender differences 233 in coping strategies across various stress levels (Viertiö et al. 2021). Moreover, this finding 234 also suggests a more multifaceted interplay of stressors influencing smoking behavior in 235 women. 236

Graves and Viertio et al reference stress and not stress and smoking. You argument here looking at stress influencing smoking behavior among women is not supported by this evidence. You are more of making assumptions. However, we need evidence to support this argument.

Page 7 lines 248 – 249

It underscores the role of diverse socio-cultural characteristics in shaping smoking behaviors.

Define “socio-248 cultural characteristics.”

Page 7 lines 249 – 251

Comparative analysis reveals that 249 social influences, psychological well-being, and cultural contexts are significant predictors 250 of smoking initiation, varying with the age and gender of youth.

Define social influences, psychological well-being, and cultural contexts

Page 7 lines 253 – 254

In essence, it elucidates that the impact of socio-environmental and 253 psychosocial factors generating these stresses varies according to gender.

As above – define

4.3. Limitations and Future Directions

Page 7 line 264 - 265

It is important to acknowledge the limitations of our study, such as the reliance on 264 self-reported data and the cross-sectional nature of the analysis.

What is the limitation of cross-sectional nature of the analysis?

Page 8 lines 266 – 267

What is wrong in focusing on young adults aged 20 – 30?

Comments on the Quality of English Language

English is good, may need some minor editing.

Round 2

Reviewer 1 Report

Comments and Suggestions for Authors

The authors have diligently revised their submission, addressing all major concerns I raised in the first round of review. I now have no further content-related feedback. Well done.

Comments on the Quality of English Language

Overall, this is a well-written manuscript. I do, however, recommend a final proofread.

For example, a study from 2021 should be written about in the past tense. 

“In 2021, the smoking rates among general adults in the continental United States are 13.1% for men, 10.1% for women, 12.9% for non-Hispanic White adults, 11.7% for non120 Hispanic Black adults, and 5.4% for non-Hispanic Asian adults (CDC 2023).”

I would also revise the following to be more concise:

The use of tobacco products other than cigarettes in young adults is known. Examples of these tobacco products include electronic cigarettes (e-cigarettes), cigarettes, cigars, smokeless tobacco, and hookah (CDC, 2022).

Perhaps as follows:

“Young adults are known to use cigarettes and other tobacco products, including electronic cigarettes (e-cigarettes), cigars, smokeless tobacco, and hookah (CDC, 2022).”

Final minor edits will ensure that the best possible version of the work is published.
